# Improvement of Quantum Approximate Optimization Algorithm for Max–Cut Problems

**DOI:** 10.3390/s22010244

**Published:** 2021-12-30

**Authors:** Javier Villalba-Diez, Ana González-Marcos, Joaquín B. Ordieres-Meré

**Affiliations:** 1Hochschule Heilbronn, Fakultät Management und Vertrieb, Campus Schwäbisch Hall, 74523 Schwäbisch Hall, Germany; javier.villalba-diez@hs-heilbronn.de; 2Complex Systems Group, Universidad Politécnica de Madrid, Av. Puerta de Hierro 2, 28040 Madrid, Spain; 3Department of Mechanical Engineering, Universidad de La Rioja, San José de Calasanz 31, 26004 Logroño, Spain; ana.gonzalez@unirioja.es; 4Escuela Técnica Superior de Ingenieros Industriales (ETSII), Universidad Politécnica de Madrid, José Gutiérrez Abascal 2, 28006 Madrid, Spain

**Keywords:** Industry 4.0, quantum approximate optimization algorithm, value–stream networks, optimization

## Abstract

The objective of this short letter is to study the optimal partitioning of value stream networks into two classes so that the number of connections between them is maximized. Such kind of problems are frequently found in the design of different systems such as communication network configuration, and industrial applications in which certain topological characteristics enhance value–stream network resilience. The main interest is to improve the Max–Cut algorithm proposed in the quantum approximate optimization approach (QAOA), looking to promote a more efficient implementation than those already published. A discussion regarding linked problems as well as further research questions are also reviewed.

## 1. Introduction

Value chains linked to Industry 4.0 (I4.0) involve complex cyber-physical networks in which information is processed efficiently by humans and machines to deliver the desired product to a customer [1,2,3]. I4.0 and the Industrial Internet of Things (IIoT) both describe further emerging landscapes for an integrated human–machine interaction [4,5]. Together, the two concepts are grounded in intelligent, interconnected cyber–physical manufacturing systems that are fully equipped and capable of controlling the process flow of industrial production. Given that many decisions are made independently by machines interoperating with production planning and fabrication systems, the integration of human users requires new paradigms [6].

In the realm of IIoT I4.0 manufacturing, I4.0 vision has advanced the notions of smart fabrication and smart factory by augmenting all assets with sensor–based connectivity [7]. These intelligent sensors generate a large amount of manufacturing data that helps to create digital twins as support for a live mirror of physical processes [8,9]. The ambition is to capture process variability within this approach, with the capability to process all relevant information by analyzing big data in cloud computation so that manufacturers are able to find bottlenecks in manufacturing processes, identify the causes and impacts of problems in such a way that the effective application of measures is useful for both product design and manufacturing engineering, including maintenance, repair and overhaul [10].

Quantum near–term simulations in classical computers have been recently used to solve different applications [6,11], including Industry 4.0 challenges such as the modelling of organizational decision networks as quantum circuits [12]. In this work, with the help of quantum simulations, a new solution for the combinatorial optimization problem is proposed, which can be applied to a wide range of applications including in the Industry 4.0 environment. It consists in finding the “optimal” partitioning of a value chain into two classes, such that the number of connections between them is maximized. Direct applications are linked to introduce flexibility in value chain models by enabling extra resilience to it, no matter whether the related processes are logistics or production related ones. Solving this industrial process design problem potentially allows maximizing the interaction between the elements of the value chain and thus maximizes its productivity [13,14]. Other applications can be connected to the Narrowband Internet of Things (NB–IoT) technology. NB–IoT is a cellular radio–based access protocol specified by 3GPP to tackle the quickly growing market for low–power wide–area connectivity significantly targeting mobile use cases. To realize the global outreach and broad adoption of NB–IoT services, mobile network operators (MNOs) need to guarantee end-to-end devices and services across several vendors connected to the deployed NB–IoT systems, and that the data transport capacity and connection modes are well understood. In this context, efficient dynamic partitions depending on the low power network available are relevant for providing a robust integrative configuration with limited transport overhead.

A solution to these problems can be formulated in terms of a combinatorial optimization approach, which involves finding an optimal object out of a finite set of objects. In this particular case, it involves finding ”optimal” bitstrings composed of 0’s and 1’s among a finite set of bitstrings. Such bitstring represents a partition of nodes of a graph into two sets, such that the number of edges between the sets is maximum. Each of the sets represents the allocation of nodes in the value stream network or nodes in the IoT system to specific managerial structures giving a maximal flexibility by providing the highest degree of connectivity.

This optimization challenge is already known as the Max–Cut problem, and it is one of the most studied combinatorial optimization problems because of its wide range of applications and because of its connections with other fields of discrete mathematics [15]. Different solutions have been proposed for the Max–Cut type of problems, as it belongs to the so-called NP-hard complexity class problems, where no known algorithms are able to solve NP-hard problems in polynomial time and thus exact methods rapidly become intractable. Such solutions include search-based algorithms [16], Machine Learning alternatives [17], as well as Recurrent Neural Networks and Reinforcement Learning [18,19].

Quantum approaches were also proposed with a quantum approximate optimization algorithm (QAOA) by [20]. The QAOA belongs to the class of hybrid algorithms and requires, in addition to the execution of shallow quantum circuits, a classical optimization process to improve the quantum circuit itself. The QAOA is an algorithm that uses unitary transformations $U(\beta_{i},\gamma_{i})$, depends on two parameters $\beta_{i}$ and $\gamma_{i}$, and is arranged in alternating blocks a number *p* of times (i∈{1,…,p}) given by
(1)$$|\psi (\vec{\beta },\vec{\gamma })\rangle =\underset{p\mathrm{times}}{\underbrace{U\left(\beta_{i}\right)U\left(\gamma_{i}\right)\ldots U\left(\beta_{i}\right)U\left(\gamma_{i}\right)}}|\psi_{0}\rangle$$
where $|\psi_{0}\rangle$ is a suitable state and parameters $\vec{\beta },\vec{\gamma }\in R^{p}$.

The goal of the algorithm is to find the combination of parameters that allows a quantum state $|\psi (\vec{\beta_{opt}},\vec{\gamma_{opt}})\rangle$ to yield the optimal solution [21]. This heuristic algorithm produces then a bit string $x\in {\left\{0,1\right\}}^{n}$ that with high probability is expected to give a good approximation of the theoretical solution. The algorithm follows a classical optimization scheme: first prepares a parameterized quantum state $|\psi (\vec{\beta },\vec{\gamma })\rangle$ (called the ansatz), then computes the parameters $\left(\vec{\beta_{opt}},\vec{\gamma_{opt}}\right)$ such that the expectation value of the quantum state is given by
(2)$$F_{p}=\langle \psi (\vec{\beta },\vec{\gamma })|H_{p}|\psi (\vec{\beta },\vec{\gamma })\rangle$$
is maximized with respect to the problem Hamiltonian $H_{p}$, and finally performs a classical optimization until some convergence criterion is reached. An overview of this is shown schematically in Figure 1.

The convergence criterion is in the Max–Cut cost function given by
(3)$$C\left(x\right)=\sum_{i,j=1}^{n}x_{i}\left(1-x_{j}\right),$$
which can be mapped to a Hamiltonian that is diagonal in the computational basis by
(4)$$H=\sum_{x\in {\left\{0,1\right\}}^{n}}^{}C\left(x\right)|x\rangle \langle x|,$$
in which $x\in {\left\{0,1\right\}}^{n}$ labels the computational basis states $|x\rangle \in C^{2^{n}}$. The expansion of $Z_{i}=\left(\begin{array}{cc} 1 & 0\\ 0 & -1 \end{array}\right)$ Pauli–Z operators can be obtained from the canonical expansion of *C*(*x*) by substitution of every variable $x_{i}\in \left\{0,1\right\}$ by the matrix $\frac{1}{2}\left(1-Z_{i}\right)$.

As indicated in the abstract, this paper aims to show that the already suggested approximate solution can be improved. The proposal for an alternative quantum algorithm configuration improves the existing solutions up to thirty nodes. The optimization algorithm proposed in Farhi et al. [20] promotes a specific sequence of unitary operators, which means an effective expression for the Hamiltonian. Finally, such a sequence of transformations will perform differently when the size of the circuit evolves. Our approach can be understood in the end as a proposal for a different sequence of unitary operators, providing a different configuration for the Hamiltonian. Then, what it is claimed is that our algorithm (our effective expression for the Hamiltonian) performs much better than the existing one.

The solution is implemented in a simulated quantum hardware environment, however there are already studies showing the time and noise effects over these algorithms when implemented in real hardware [22].

We structure the rest of the work hereinafter as follows: Section 2 outlines the modified architecture in a reasoned manner. Then, Section 3 presents the results of the algorithm as compared with the analytical solution, which for when $|\psi (\vec{\beta_{opt}},\vec{\gamma_{opt}})\rangle$ is not too deep can be computed classically, and the results previously obtained by [20]. Finally, Section 4 briefly discuss the obtained results, outlines future lines of research, and presents limitations in the presented work.

## 2. Modified QAOA

In this section, we present the results of the algorithm applied to a value stream network of *n* = 10 nodes. The complete results for other configurations are available in open access in [23].

We start by representing in Figure 2 the value stream network as a graph $G=\left\{n,e\right\}$ of *n* = 10 unlabeled nodes and *e* = 13 edges.

If the graph coincides with the connectivity of our logical network (either IoT topology or value stream network), the cost function C(x) coincides with the hamiltonian *H* used to generate the state.

For a shallow approximation with *p* = 1, the analytical solution for the expectation value is given by
(5)$$F_{1}\left(\beta ,\gamma \right)=\langle \psi_{1}\left|H\right|\psi_{1}\rangle$$

Combining Equations (3) and (4), the Hamiltonian *H* makes use of the expectation value to measure the edges individually:(6)$$f_{i,k}\left(\gamma ,\beta \right)=\frac{1}{2}\langle \psi_{1}\left(\gamma ,\beta \right)\left|\left(1-Z_{i}Z_{k}\right)\right|\psi_{1}\left(\gamma ,\beta \right)\rangle$$

There are two types of edges: those that connect a node with degree one (A), and those that connect a node with degree three (B). For the A–class edges, an example of the encoding of the optimization function between nodes (0) and (1) is given by
(7)$$2f_{A}=1-\langle {+}^{1}\left|U_{01}\left(\gamma \right)U_{12}\left(\gamma \right)U_{13}\left(\gamma \right)X_{0}\left(\beta \right)X_{1}\left(\beta \right)Z_{0}Z_{1}X_{1}^{†}\left(\beta \right)X_{0}^{†}\left(\beta \right)U_{01}^{†}\left(\gamma \right)U_{12}^{†}\left(\gamma \right)U_{13}^{†}\left(\gamma \right)\right|{+}^{1}\rangle$$
and for the B–class edges, the encoding of the optimization function between nodes (1) and (2) is given by
(8)$$2f_{B}=1-\langle {+}^{3}|U_{21}\left(\gamma \right)U_{24}\left(\gamma \right)U_{23}\left(\gamma \right)X_{1}\left(\beta \right)X_{2}\left(\beta \right)Z_{1}Z_{2}X_{1}^{†}\left(\beta \right)X_{2}^{†}\left(\beta \right)\left(\gamma \right)U_{12}^{†}\left(\gamma \right)U_{23}^{†}\left(\gamma \right)U_{24}^{†}\left(\gamma \right)|{+}^{3}\rangle$$
in which $|{+}^{n}\rangle =\sum_{x\in {\left\{0,1\right\}}^{n}}^{}\frac{1}{\sqrt{2^{n}}}|x\rangle$ prepares for an equal superposition state followed by a sequence of parametrized unitary operations. As shown in Equations (7) and (8) these unitary operations are a combination of parametrized Hamiltonian cost $U_{C}\left(\gamma \right)=e^{-i\gamma H_{C}}$ and mixer layers $U_{M}\left(\beta \right)=e^{-i\beta H_{M}}$. The subindexes in the unitary operations indicate the nodes on which the operators act upon.

In our case *n* = 10, there are two A–class edges and eleven B–class edges. This yields Equation (9), which is depicted in Figure 3 which shows the periodicity in both parameters and exhibits a highly nonlinear behaviour.Farhi et al. [20] proposed QAOA with the structure represented in Figure 4.
(9)$$\begin{array}{c} F_{1}\left(\beta ,\gamma \right)=2f_{A}\left(\beta ,\gamma \right)+11f_{B}\left(\beta ,\gamma \right)=[\sin\left(4\gamma \right)\sin\left(4\beta \right)+\sin^{2}\left(2\beta \right)\sin^{2}\left(2\gamma \right)]+\\ \frac{11}{2}[1-\sin^{2}(2\beta )\sin^{2}(2\gamma )\cos^{2}(4\gamma )-\frac{1}{4}\sin(4\beta )\cos(4\gamma )(1+\cos^{2}(4\gamma ))] \end{array}$$

Such an algorithm starts by preparing the system in superposition with a Hadamard gate on all qubits. Next, a rotation of 2γ is applied to each of the edges if both are in state |11〉. This conditional rotation has the form given by
(10)$$C_{p}\left(2\gamma \right)=\left(\begin{array}{cccc} 1 & 0 & 0 & 0\\ 0 & 1 & 0 & 0\\ 0 & 0 & 1 & 0\\ 0 & 0 & 0 & e^{-2i\gamma } \end{array}\right).$$

This allows the algorithm to be applied when both qubits are in state |1〉 simultaneously. A quantum phase correction of γ is then applied to each of the nodes joined by each edge. This rotation has the form given by
(11)$$p\left(\gamma \right)=\left(\begin{array}{cc} 1 & 0\\ 0 & e^{i\gamma } \end{array}\right).$$

Such configuration allows the previous rotation to be neutralized when the two qubits are in state |11〉. The result of these two operations allows a rotation of γ to be applied to all nodes as long as both communicating nodes are not simultaneously in state |1〉.

Finally, a rotation around the *X*-axis of 2β, perpendicular to the computing axis, is applied to all nodes. This rotation has the form given by
(12)$$R_{x}\left(2\beta \right)=\left(\begin{array}{cc} \cos\beta & -i\sin\beta \\ -i\sin(2\beta ) & \cos(2\beta ) \end{array}\right).$$

In summary, in [20] the QAOA algorithm applies, after a standard superposition, a quantum phase of γ to every node connected to each other, as long as both are not in state |11〉, and a rotation around the perpendicular to the computational axis of 2β to all the nodes.

On the other hand, this paper proposes a novel QAOA approach represented in Figure 5.

Analogous to the previous example, our algorithm starts by preparing the system in superposition with a Hadamard gate on all qubits. We then perform a conditional rotation of γ to each node connected to another if the second is in state |1〉 in both directions. This is done by concatenation of two $U3\left(\frac{\gamma }{2},0,0\right)$ and $U3\left(\frac{-\gamma }{2},0,0\right)$ gates given by
(13)$$U3\left(\frac{\gamma }{2},0,0\right)=\left(\begin{array}{cc} \cos\left(\frac{\gamma }{4}\right) & -\sin\left(\frac{\gamma }{4}\right)\\ \sin\left(\frac{\gamma }{4}\right) & \cos\left(\frac{\gamma }{4}\right) \end{array}\right),$$
and a conditional *CX* rotation applied to one of the nodes $q_{0}$ taking the other $q_{1}$ as control given by
(14)$$CX_{q_{0},q_{1}}=\left(\begin{array}{cccc} 1 & 0 & 0 & 0\\ 0 & 0 & 0 & 1\\ 0 & 0 & 1 & 0\\ 0 & 1 & 0 & 0 \end{array}\right).$$

This method works because when the control *qubit*
$|\Psi_{0}\rangle$ is in state |0〉, all we have is $U3\left(\frac{\gamma }{2},0,0\right)$ followed by a $U3\left(\frac{-\gamma }{2},0,0\right)$ and the effect is trivial. On the other hand, when the control *qubit*
$|\Psi_{0}\rangle$ is in state |1〉, the net effect is a controlled rotation U3(γ,0,0) on the $|\Psi_{1}\rangle$
*qubit*. These rotations are taken in both directions because our network is not directed. This algorithm is expected to yield better results than the previous one because the transformations are differential as a function of node state.

Finally, as in the previous algorithm, a rotation around the *X*-axis of 2β, perpendicular to the computing axis, is applied to all nodes. This rotation has the form given by Equation (12).

## 3. Results

In this section, we present the results of the algorithm applied to a logical nondirected network of *n* = 10 nodes. The quantum simulations presented were simulated on *qiskit* tool, a Python–based quantum computing platform developed by IBM [24], and the code and additional results can be accessed in this Open Access Repository: [23].

The results confirm our expectations and our proposed QAOA algorithm predicts the analytical results better for a shallow quantum circuit with p=1. A summary of the results for different numbers of nodes is shown in Figure 6. In Table 1 we represent the comparison of the analytical solution curve and the respective QAOA algorithms. Our solution shows better performance in all metrics.

The bit string that delivers the optimal solution is x={0110011010}, as shown in Figure 7. This graph clearly shows the configuration obtained by the QAOA algorithm presented with two types of nodes represented in two colors, green {0} and blue {1}.

## 4. Discussion, Future Lines of Research, and Limitations

The analysis of the solution presented in Table 1 shows to what end the quality of the solution found improves the previous one, which justifies the spent effort in considering smarter quantum circuits for the operation of the QAOA algorithm since there is not exist a universal strategy that works across a broad range of optimization problems. Based on the proposal made, the benefit of the algorithm proposed is evident against other existing algorithms, at least in the case of *p* = 1. As a consequence, value–stream network design challenges can be better understood with the aid of this quantum optimization algorithm. More research is needed to analyze the evolution of potential benefits when the number of transformation blocks grows up. Indeed, resources and performance figures are also needed to get the whole perspective.

The network topology can be modified, however, if classes for the number of links per node are extended, then a new formulation for the cost function introduced in Equation (9) is required.

Future research lines will involve the implementation of this new proposal for quantum circuits in physical quantum computers, to analyze both the performance and the stability against noise, not only for *p* = 1 but also when the transformation blocks are increased. Moreover, since two-qubit gates (e.g., CNOT gates) are significantly more erroneous than single qubit gates, the proposal of smarter circuits with reduced number of two-qubit gates, such as the optimization proposed by [22], is an area of interest.

Because of the problem formulation, the network was defined and just the link of nodes with different managerial classes was the goal. However, it could be possible to reverse the problem and start from the node type distribution and look to connect those nodes with a number of edges optimizing the imputation rules between them.

Some limitations can be found regarding the applicability to real cases, because the existence of extra constraints applicable to nodes or edges. Therefore, additional aspects related to penalty terms when formulation of the C(x) function could be a potential workaround. Following this line, another relevant research area is to extend the current formulation for the Max–Cut problem to the Max–k–Cut one, in line with the recent analysis provided by [26]. Although we have obtained satisfactory results with p=1, we can expect a better approximation for a larger number of qubits if we increase the *p* parameter. This would, however, entail additional relative difficulties in factoring the Hamiltonian in the adiabatic hypothesis that may be problematic in practice.

## Figures and Tables

**Figure 1 sensors-22-00244-f001:**
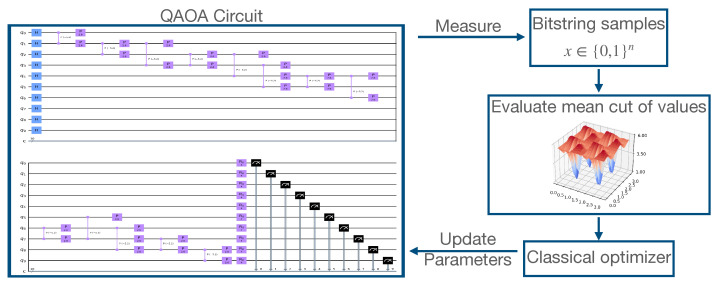
QAOA Overview.

**Figure 2 sensors-22-00244-f002:**
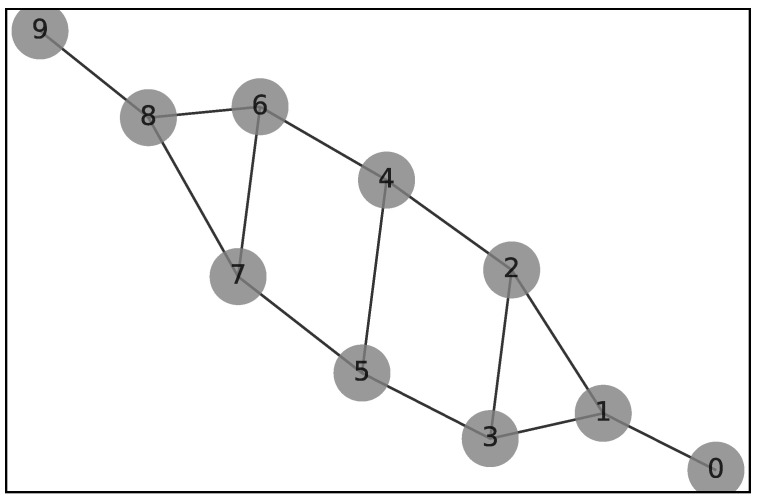
Value stream network with *n* = 10 nodes.

**Figure 3 sensors-22-00244-f003:**
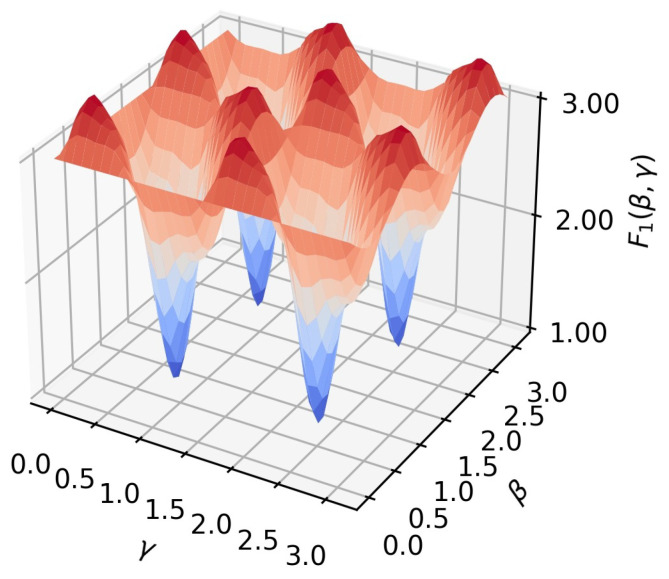
Analytic solution for *p* = 1 and value stream network configuration from Figure 2.

**Figure 4 sensors-22-00244-f004:**
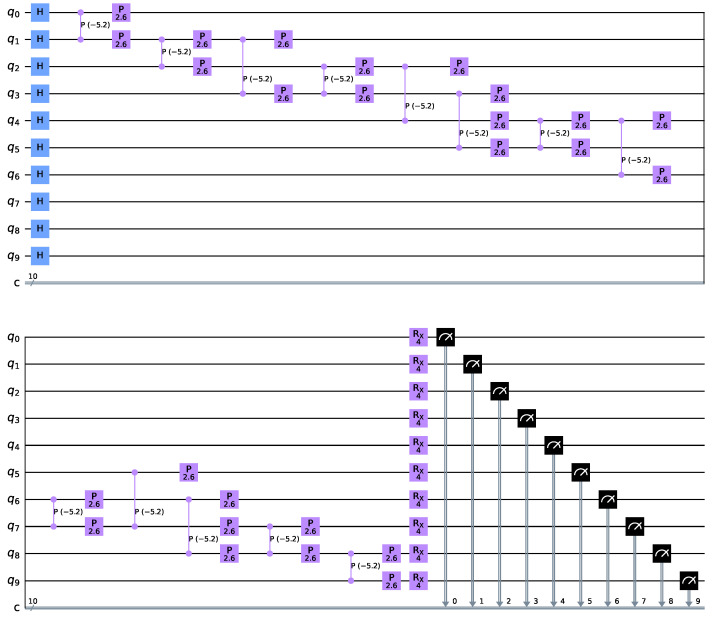
QAOA—Farhi et al. [20].

**Figure 5 sensors-22-00244-f005:**
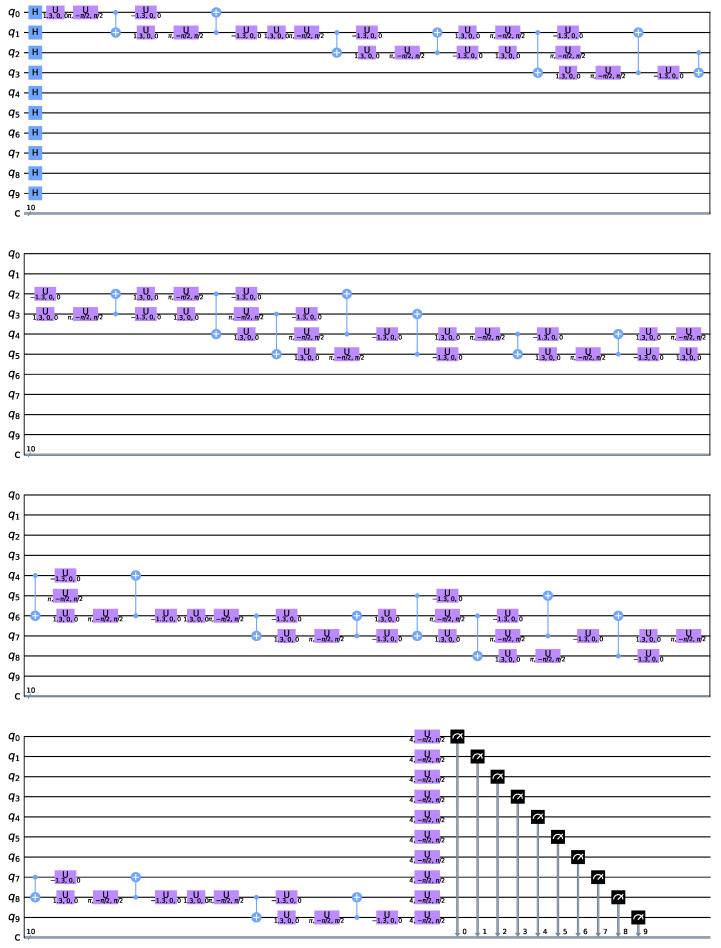
QAOA—Villalba–Diez et al.

**Figure 6 sensors-22-00244-f006:**
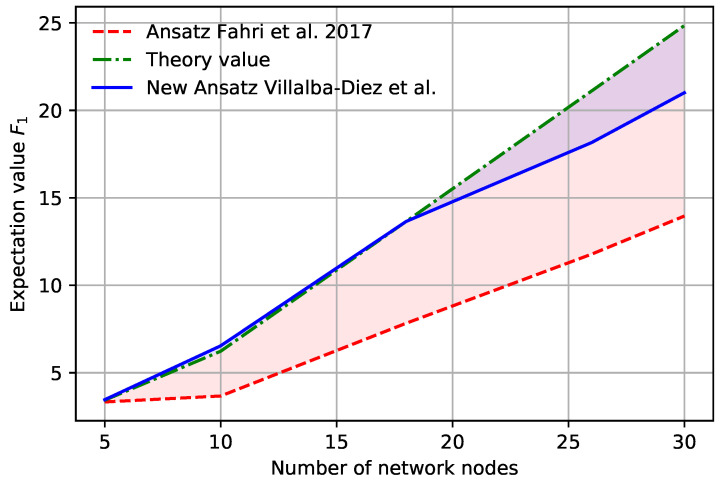
QAOA results comparison.

**Figure 7 sensors-22-00244-f007:**
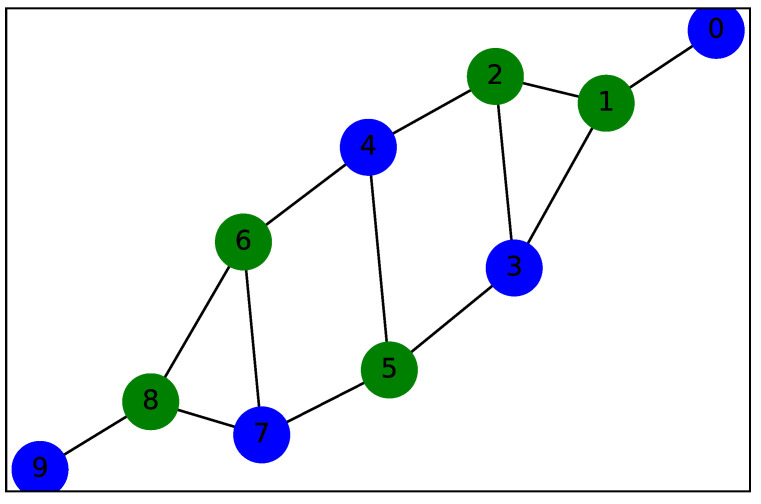
Value stream network clustering with QAOA.

**Table 1 sensors-22-00244-t001:** Results comparison for different measures for identifying curve similarity [25].

	Analytic vs.	
	**Farhi et al.**	**Villalba et al.**
Directed Hausdorff distance	8.22	3.84
Discrete Fréchet distance	10.89	3.84
Dynamic Time Wrapping	28.70	7.13
Partial Curve Mapping	1.6893	0.3223
Area between two curves	1.2744	0.3642
Curve-Length distance metric	141.21	26.23

## Data Availability

The code can be found at https://shorturl.at/hnpC3 (accessed on 1 December 2021) [23].

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
