# Peer review of "Improvement of Quantum Approximate Optimization Algorithm for Max–Cut Problems"

_sensors, 2021, doi:10.3390/s22010244_

Round 1
Reviewer 1 Report
In the paper “Improvement of Quantum Approximate Optimization Algorithm for Max–Cut Problems” , Villalba-Diez et al. introduce a modification for the QAOA algorithm and test its performance within the Max-Cut problem in a particular graph.
The topic is interesting and timely. It is crucial determine if quantum algorithms present any kind of advantage over their classical counterparts in AI or optimization tasks. In fact, improving QAOA is very important.
On the other hand, the document needs some improvements to be able to be published.
-The presentation. Is it necessary to plot the full circuit? In the current format the circuit takes up a lot of space, and since all the qubits are shown the different gates are difficult to read in print. Besides, the authors may consider rewriting some sentences that are difficult to read as “All metrics show better performance of our novel solution” or “This goal of this brief letter…”
“
Quantum near–term simulations in classical computers have been recently used to solve different 30 applications [11,12], including Industry 4.0 challenges [13–17]. In this work, with the help of quantum simulations, a new solution for the combinatorial optimization problem is proposed, which can be 32 applied to a wide range of applications including in the Industry 4.0 environment.
“
Do you mean that your results are obtained using a simulator? I.e. they are not run in a q-computer. Are you using skit? In that case, you should cite it.
-The results:
* The QAOA structure can be trace back to the adiabatic algorithm. If $p\to \infty$, the QAOA the evolution “prepares” the g.s. of Hp (i.e. the solution) I.e. the ansatz is justified. In the proposed modification there is no intuition why it will work. It would help to describe the Unitary explicitly, in addition to the proposed circuit.
*The explicit Unitary is important since a circuit can be re-organized and simplified such that different arrangement of gates provide equivalent unitaries.
*The improvement is only tested in one particular graph and for $p=1$. More tests are neccesary. Its better performance could be very particular from the p=1 case considered and/or the graph.
Doubt:
-If I am correct U3(+\gamma/2,0,0)^\dagger= U3 (-\gamma/2, 0, 0) . Thus, U3(+\gamma/2,0,0) U3 (-\gamma/2, 0, 0) =1 always. Then, I do not understand why its application is different applied to either 0 or 1 (following your discussion below Eq. 8)
If the authors address this point, the article could be published in Sensors.
Reviewer 2 Report
Report is attached as a pdf file.

Author Response
We want to thank the very professional work and careful analysis you did over our draft letter. We also thanks for all the valuable suggestions provided.
We were addressing all the comments and suggestions you have raised, including the minor / editorial concerns.
There are other relevant aspects raised in section 2 of your review. First of all, it's worth to mention that our work is not a full paper but a Communication paper (kind of Short letter). It is relevant as potentially readers will not expect a full development of the theoretical background. This is relevant as the reviewer is asking for more references, but it works against the limitations imposed by the journal because of the adopted format.
We have removed the self-citation, avoiding any kind of confusion or bias in the motivation for our work.
Indeed, the arXiv references where replaced by peer-reviewed publications.
More explanations have been introduced to cope with vector parameters in such a way it becomes consistent with the notation.
More details were provided for the proposed algorithm with the introduction of the relevant gates, highlighting differences with previous algorithm. The cost function was kept to visualize the periodicity properties and its high non-linearity structure. However, a better explanation for the structure of the function itself was provided
Deeper discussion for the impact of the algorithm, as well as side connections deserve a longer work, currently under preparation, which require a full paper. Our intention here is just to emphasize the value of an improvement for the optimization algorithm itself, therefore we have selected a letter style (short document focused on presenting narrow contributions but potentially with significant relevance).
We hope the second version of our communication provides higher understanding and, correspondingly more value for readers.
Reviewer 3 Report
The article is well written and contains a lot of useful information for a letter. On the other hand, I wonder if it fits into "Sensors". Perhaps "Applied Sciences, MDPI" is better, or even "Entropy". Overall, it is a good solid piece of work presenting a smart idea! Small corrections/modifications are required:
1) Please show the algorithm as a bulleted algorithmic procedure (step 1, step2 etc.), hence making the quantum circuit easier to read (as shown in fig 3) (if you can modify the figure adding "{ ...]" corresponding to the gates application and the stepwise procedure would be even better).
2) The explanation provided with equations 7 and 8 is very well understood by an expert but not by a newcomer in the field. I suggest that part to be expanded and include an extra paragraph/math type where the phrases "This method works.... node state" are analyzed in a more "mathematical" way rather than just mentioning the final outcome.
Author Response
Thank you for the effort in reading the draft of this letter and thank you for your positive comments.
Regarding your first comment about fitness in Sensors, we understand your comment, however we think there is enough room for such kind of papers getting IoT and IIoT devices being managed through new optimization techniques. Indeed quantum applications will become more and more popular, so we think our decision to keep Sensors as the first target is still meaningful.
Regarding your second comment regarding the more detailed steps for the algorithms, a better explanation was provided both, for the original Farhi et al. algorithm as well as for our proposal. It does include Fig 1 to bring additional visual scaffolding.
Finally, regarding your third comment, a simplified version of the formulation involving single A and B type nodes have been considered, where more details about the involved components have been detailed.
We hope now it becomes more readable, and we want to thank you for the suggestions you have brought.
J. Ordieres on behalf of the authors
Round 2
Reviewer 1 Report
The authors have partially addressed my questions.
I still don't see how your modified algorithm relates to the unitaries in Eq. (1). I assume they are not implementing the same unitaries as in Farhi (so they are implementing different effective Hamiltonians).
It would be nice to have labels in figure 3.
Author Response
Thank you very much for your comments.
Yes, you are fully right. Actually we have included your perspective as a way to interpret the meaning of the proposed algorithm in lines 83-89.
Again many thanks for the time you have invested in adding value to our paper.
Joaquín, on behalf of the authors.
Reviewer 2 Report
The file is attached

Author Response
Thank you very much for the time you have spent in processing our paper and providing valuable feedback. We, the authors, really thank you because of it.
Regarding your comments, the first one was a typo, coming from the copy and paste for the right hand side of the equation. As you said the product had no sense as it is just Fp
To introduce Psi_{p} creates much more confussion than benefit as it tries to represent the optimal configuration, therefore we decided to remove it and keep the subindex opt.
Regarding the points 2 and 3 we have fixed the issues according to your suggestions, which were absolutely right.
Again, many thanks indeed
Joaquín, on behalf of the authors.